# Harnessing Marine Algae for Sustainable Agriculture: Natural Bioactive Compounds as Eco-Friendly Pesticidal Agents

**DOI:** 10.3390/md23090346

**Published:** 2025-08-28

**Authors:** Georgi Beev, Diyana Dermendzhieva, Zvezdelina Yaneva, Georgi Kalaydzhiev, Nikolina Naydenova, Daniela Stoeva, Denitsa Georgieva, Silviya Hristova, Zornitsa Beeva, Nikolay Petrov

**Affiliations:** 1Department of Biological Sciences, Faculty of Agriculture, Trakia University, Students Campus, 6000 Stara Zagora, Bulgaria; daniela.stoeva@trakia-uni.bg (D.S.); nikolay.d.petrov@trakia-uni.bg (N.P.); 2Department of Applied Ecology and Animal Hygiene, Faculty of Agriculture, Trakia University, 6000 Stara Zagora, Bulgaria; diyana.dermendzhieva@trakia-uni.bg; 3Department of Pharmacology, Animal Physiology, Biochemistry and Chemistry, Faculty of Veterinary Medicine, Trakia University, Students Campus, 6000 Stara Zagora, Bulgaria; zvezdelina.yaneva@trakia-uni.bg (Z.Y.); denitsa.georgieva@trakia-uni.bg (D.G.); silviya.hristova@trakia-uni.bg (S.H.); 4Department of Livestock-Ruminants and Technologies of Animal Products, Faculty of Agriculture, Trakia University, Students Campus, 6000 Stara Zagora, Bulgaria; georgi.kalaydzhiev@trakia-uni.bg (G.K.); nikolina.zheleva@trakia-uni.bg (N.N.)

**Keywords:** marine algae, crop resilience, sustainable pest control, algae-based products, alternative pesticides, low-input farming

## Abstract

Currently, marine algae are capturing the attention of both farmers and researchers eager to integrate sustainable methods to safeguard their crops. Instead of relying exclusively on synthetic pesticides, which often have negative environmental effects, some growers are now exploring algae-based products in hopes of reducing pest pressures. Various natural compounds sourced from algae—such as specific fatty acids and complex sugars—are believed to inhibit pest development, although their precise mechanisms are yet to be fully understood. Furthermore, there is some evidence suggesting that these compounds may bolster the plant’s own immune responses, thus enhancing crop resilience. Despite certain limitations on field applications, various techniques, including spraying, amending soil, or pre-treating seeds, are currently being evaluated. The results from the laboratory present a positive outlook, but implementing these discoveries to ensure consistent efficacy in practical settings is a major challenge. Variables such as climatic fluctuations, product durability, and formulation standards all elevate this complexity. In every instance, the approach of incorporating algae to lessen chemical dependence while securing uniform yields persists in being of interest, particularly in the area of organic or low-input farming.

## 1. Introduction

Marine algae, comprising various important seaweeds and smaller organisms, are receiving growing academic interest for their possible function in eco-conscious pest management solutions [1,2,3]. Numerous chemical components are generated by these biological systems, which include phenolics, terpenoids, sulfated polysaccharides, fatty acids, and alkaloids based on nitrogen [4,5,6]. These specific metabolites assist algae in adjusting to life in marine settings, while increasing evidence shows they could also potentially affect agricultural pests that harm crop success. Many natural substances closely resemble the activity of conventional chemical agents such as antibiotics and insecticides, and they can effectively suppress microbial growth and deter pests [2,7]. With their natural origin, algal biopesticides offer a more sustainable and safer alternative to synthetic pesticides, whose residues often persist and rank among the most significant environmental pollutants, posing long-term risks to both the nature and human health [8,9]. Conversely, organic substances from marine algae usually degrade quickly and are less harmful to non-target species [3,5]. Laboratory studies show that these compounds may suppress pest growth and reproduction, offering protection for beneficial species like pollinators [10,11]. Not only limited to pest control, certain algae extracts appear to spark the defensive responses in plants, thereby increasing the crops’ resistance to diseases and supporting their overall development and robustness [12,13,14]. Employing marine algae in compost or as soil supplements yields substantial agricultural benefits by nurturing microbial diversity and renewing essential nutrients found in the ground [15]. However, ongoing examinations are vital to fully realize the lasting environmental consequences associated with these biopesticides. Uncertainties persist regarding the potential impacts of repeated applications on aquatic systems or the possible unintended alterations in ecological equilibrium [16,17].

This review explores the role of marine algae in sustainable agriculture by examining their pesticidal properties, explaining how these compounds work and are applied, and highlighting the ecological and practical advantages they offer over chemical pesticides.

## 2. Pesticidal Properties of Marine Algae

### 2.1. Insecticidal Activity

Chemical resistance is a major issue, and more people are looking to nature for safer options. Marine algae have proven effective in controlling various agricultural pests, including insects and mites, by targeting biochemical pathways, including enzymatic activity, physiological stability and development, positioning them as strong candidates for safer, greener pest control methods [18,19]. Researchers have discovered that algae extracts can interrupt the insect life cycle by stopping eggs from hatching, killing larvae, or slowing their development as they grow. What is exciting is that these effects are not limited to a single pest—they have been observed in both mosquitoes and crop-damaging insects [20,21,22,23]. One advantage is that these natural products break down fast and do not stick around to harm helpful animals like bees or earthworms [3,24]. One clear example is *Sargassum latifolium*, a brown algae species whose compounds block the chitin production, a key material in an insect’s outer shell. Without enough chitin, insects cannot develop properly—or survive at all [25]. The species *Padina pavonica* (Phaeophyceae) also shows good insecticidal activity. Its benzene extract was tested on a cotton bug (*Dysdercus cingulatus*), and the results were strong: fewer eggs hatched, more young insects died, and internal damage was clear from protein and DNA loss [26]. Other brown algae, like *Sargassum wightii* and *Turbinaria ornata*, produce tannins that interfere with enzymes insects need—such as amylase and protease—to properly digest their food. For instance, Petchidurai et al. [11] reported that the cotton leafhopper (*Amrasca devastans*) could not digest food properly, which led to increased mortality. A separate study found that methanolic extracts of *S. wightii* killed over 83% of diamondback moth (*Plutella xylostella*) larvae at a 3% dose [27]. Detailed profiling of *S. wightii* revealed the presence of ethyl salicylate, methyl salicylate, ethyl palmitate, palmitic acid, oleic acid, phytolin, and diethyl phthalate, which are responsible for its larvicidal efficacy against mosquitoes and moth larvae [28]. Green algae demonstrated their insecticidal potential too. Extracts from *Ulva lactuca* and *Caulerpa scalpelliformis* (Chlorophyta) proved toxic to *Spodoptera litura* larvae, with higher doses leading to higher death rates [29]. In particular, *Chaetomorpha antennina* (Chlorophyta) produces fatty acids such as hexadecanoic, octadecanoic, and linoleic acids, which trigger systemic defense responses in tomato plants against *S. litura* [6].

Marine algae and their metabolites are not only effective during crop growth but also show considerable promise for controlling pests in stored commodities. For example, volatile oils extracted from *Actinotrichia fragilis*, *Liagora ceranoides* (Rhodophyta), and *Colpomenia sinuosa* (Phaeophyceae) caused between 55 and 90% mortality of *Oryzaephilus mercator* and 60–80% mortality of *Tribolium castaneum* when applied at 12 μL/L air for 48 h. In particular, the oil from *A. fragilis*, which contains 39.6% 1-dodecanol, produced 80–90% mortality in both pest species [30]. In a similar vein, n-hexane extracts from the *S. latifolium* were highly effective against multiple stored-product pests, including *T. castaneum*. These extracts not only damaged the insect exoskeleton but also likely acted as chitin synthase inhibitors, as suggested by molecular docking studies [25]. Together, these results indicate that marine algal bioactive compounds could serve as eco-friendly alternatives for post-harvest pest management, complementing their established role during crop growth.

As detailed in Table 1, a variety of marine algae exhibit potent insecticidal properties, with effects ranging from enzyme inhibition to structural damage in target pests.

Even with these successes, we are still figuring out a lot. Most research uses crude extracts without isolating the specific compounds that actually work. Moreover, most tests happen in labs, not real farms. That is a problem, since the chemical makeup of algae can change with the season or where they are collected [17,41]. Different lab methods and target pests also make it tough to compare findings [18,42]. We still do not have firm guidelines or long-term safety studies, and that slows down product development [14,24]. To move this research ahead, we need to identify the most effective compounds, test them in actual farming conditions, and make sure the products work reliably, no matter how the environment changes. Marine algae might not be a perfect solution yet, but they hold real promise as safer, natural tools for modern insect control [6,39,43].

### 2.2. Fungicidal Activity

Marine algae offer promising natural alternatives for managing fungal diseases in agriculture. Researchers have studied many species across green, brown, and red algal groups which can release compounds that either suppress fungal growth or help plants strengthen their own defenses [44,45,46,47].

*Ulva lactuca* (formerly *Ulva fasciata*) and *Ulva flexuosa* (formerly *Enteromorpha flexuosa*), two green algae, reduced the growth of soil fungi like *Fusarium solani* and *Macrophomina phaseolina* in cucumber. Chloroform extracts of *U. lactuca* (formerly *U. fasciata*) cut fungal proliferation by nearly 69% [48]. The red seaweed *Solieria robusta*, when applied to soil, lowered root rot levels in soybean caused by *Fusarium solani* and also suppressed nematodes [49]. Extracts from *Laurencia johnstonii* (Rhodophyta) and *Dictyota dichotoma* (Phaeophyceae), representing red and brown algae, inhibited growth of *Fusarium oxysporum* and other pathogens [44,50]. Other brown species like *Cystoseira humilis* var. *myriophylloides* (formerly *Cystoseira myriophylloides*) and *Laminaria digitata* (Phaeophyceae) limited the spread of *Verticillium dahliae* and *Botrytis cinerea* [51]. In addition to direct antifungal action, certain algae activate plant immune systems. Foliar treatments using *Padina pavonica*, *Acanthophora spicifera*, and *Ulva lactuca* reduced the severity of rice blast caused by *Pyricularia oryzae*. Treated plants produced more peroxidase and phenylalanine ammonia-lyase, enzymes linked to disease resistance [45,52]. Beyond crude extracts, isolated compounds such as norcycloartane-type triterpenes from *Tydemania expeditionis* have shown strong inhibition against *Fusarium oxysporum* and *Colletotrichum lagenarium*, respectively [53]. Similarly, formulations derived from *Ascophyllum nodosum* (Phaeophyceae) enhanced chitinase and β-1,3-glucanase in cucumber, leading to reduced damage from *Fusarium oxysporum* and *Alternaria cucumerinum* [12]. Polyphenolic compounds also contribute: phlorotannins from brown algae exert broad antifungal effects [54], while rosmarinic acid and quercetin from *Gracilariopsis persica* inhibited *Botrytis cinerea* and *Penicillium expansum* [55]. In addition, the brown alga *Turbinaria conoides* produces a range of compounds, including sterols, fucoidans, and esters such as dibutyl phthalate, demonstrating suppressive potential on *Fusarium oxysporum* [56]. These findings highlight the value of identifying and characterizing individual metabolites to complement studies relying only on crude extract evaluations.

Green algae like *Caulerpa racemosa* and *Cladophora glomerata* generate bioactive compounds that suppress the proliferation of various phytopathogenic fungi (with documented effects against *Botrytis* and *Curvularia* spp.) [50]. Meanwhile, *Auxenochlorella protothecoides* (formerly *Chlorella protothecoides*), a well-known freshwater microalga (some strains can tolerate brackish conditions), suppressed the growth of postharvest fungi, including *Aspergillus niger* and *Penicillium expansum*, which are also major contributors to fruit and vegetable spoilage [57].

Additionally, polysaccharides—such as laminarin and carrageenan (except of their insecticidal property) act by enhancing the plant’s defense systems rather than attacking the fungi themselves [58]. This multi-target action also helps prevent resistance buildup in pathogens [47]. Table 2 summarizes the diverse antifungal effects observed across green, brown, and red algal species, highlighting both direct pathogen suppression and induced plant immunity.

Still, several obstacles remain. Algal chemistry can shift with the season, location, or harvesting method, which affects consistency [17,64]. Many studies use crude extracts without identifying which substances do the work [42]. Most experiments stay in the lab or greenhouse, so we still lack large-scale field data [18]. Some extracts degrade too fast unless stabilized [24]. In many regions, product approval is delayed by unclear regulatory systems [14].

### 2.3. Nematicidal Activity

Root-knot nematodes are a major threat in agriculture. By damaging roots and causing galls, they weaken the plant and often lead to major losses in crop yield. Scientists have started exploring marine algae as a safer, natural way to fight these pests thanks to the bioactive compounds they produce [41,65,66]. From powders to extracts, algae-based treatments have proven effective at managing nematodes. While most studies employ crude extracts, specific compounds have been identified. GC–MS analysis of the most effective fraction of *Colpomenia sinuosa* (Phaeophyceae) methylene chloride eluent revealed five major bioactive constituents: dibutyl phthalate, methyl methyltetradecanoate, palmitic acid, 1-propene-1,2,3-tricarboxylic acid and tributyl ester [67]. Using the same assay, silver nanoparticles synthesized from *C. sinuosa* and *Corallina mediterranea* (Rhodophyta, Corallinaceae) demonstrated potent nematicidal activity against *Meloidogyne incognita* in tomato plants [67], highlighting the potential of both isolated compounds and nanoparticle formulations for nematode management.

Green algae *Ulva lactuca* lowered gall formation and nematode counts in banana crops—likely thanks to its high phenolic content [68]. Published data from 2009 and 2011 reported that applying dry powdered seaweeds to nematode-infested soils or tomato plant roots led to significant reductions in nematode populations and enhanced plant biomass. The seaweeds included green (*Halimeda tuna*) (Chlorophyta), brown (*Spatoglossum asperum*, *Sargassum swartzii*, *Spatoglossum variabile*) (Phaeophyceae), and red (*Melanothamnus afaqhusainii*) (Rhodophyta) species, indicating both protective and growth-promoting effects. However, maximum nematicidal effect was observed when seaweed applications were combined with the synthetic nematicide carbofuran [49,69]. This outcome suggests that while seaweeds offer notable nematicidal potential, their effectiveness may be limited on their own, and successful nematode management still requires support from chemical agents.

Likewise, methanolic extracts and commercially available seaweed products from brown algae species *Phacelocarpus tristichus, Turbinaria ornata, Ascophyllum nodosum* and *Ecklonia maxima* lowered the number of *Meloidogyne incognita*, *Meloidogyne chitwoodi* and *Meloidogyne hapla* in tomato roots, while also helping the plants grow better [4,70]. Ethanolic extracts of *Padina concrescens* (Phaeophyceae) and *Laurencia johnstonii* also stopped a large portion of nematode eggs from hatching and reduced the survival of juveniles. Meanwhile, soil treatments using *Spatoglossum schroederi* (Phaeophyceae) cut down root galling and improved the root health of infected tomatoes [41]. An overview of nematicidal outcomes, presented in Table 3, illustrates the capacity of algal extracts to reduce root-knot nematode lations while supporting plant vigor.

The interesting part is that algae do not just go after the nematodes—they also make the plants better at protecting themselves. In particular, foliar sprays made from marine algae increased enzyme activity linked to plant resistance—such as polyphenol oxidase and peroxidase—which likely makes the plants tougher against nematode attacks [47].

Using algae and Cyanobacteria to manage nematodes comes with similar limitations as other algae-based pest controls, including differences in effectiveness depending on the environment and how the products are prepared.

### 2.4. Herbicidal Activity

Studies on the possible use of seaweed extracts as novel bioherbicide sources is a relatively new area with very limited data in this space (Table 4). Researchers have started paying closer attention to marine algae as a potential source of natural herbicides, thanks to the wide variety of biologically active compounds they produce [73]. Phenolic compounds, which are usually associated with antioxidant activity [74], have also been linked to plant growth suppression in recent studies [5,65,75,76]. That said, their role in herbicidal activity is not completely clear. For instance, Chukwuma et al. [77] tested red seaweed (*Mastocarpus stellatus* and *Porphyra dioica*) (Rhodophyta) extracts and noted that while methanol and petroleum ether extracts—rich in phenolics—were inhibitory, similar extracts made using n-hexane were not, even though they also contained phenolic compounds. This suggests that other chemicals, or combinations of several, might be doing the heavy lifting.

Another study by Haniffa et al. [78] tested how 16 different seaweed extracts affected lettuce seed growth. Seven of them had a noticeable impact at 1000 ppm. Notably, *Laurencia heteroclada* (Rhodophyta), *Caulerpa racemosa*, and *Caulerpa sertularioides* (Chlorophyta) were the most effective in suppressing seed germination and root development. Lettuce seeds treated with *C. racemosa* developed roots less than a centimeter long, while untreated seeds grew roots over three times that length. Although chemical analysis of the *L. heteroclada* extract revealed several noteworthy compounds, including algoane and caulerpin, algoane alone did not replicate the extract’s full inhibitory effect.

This finding suggests that no single compound is solely responsible for the phytotoxic activity. Instead, it points to a likely synergistic interaction among multiple components within the extract [78,79]. As shown in Table 4, several marine algae demonstrate phytotoxic effects on weed species, indicating their promise as environmentally friendly bioherbicides.

**Table 4 marinedrugs-23-00346-t004:** Herbicidal Activity of Marine Algae.

Algal Species	Target Pest	Product/Effects	Reference
Chlorophyta (green microalgae)	Model organisms	Fatty acids, aldehydes exert herbicidal effects	[5]
Microalgae allelochemicals	Weeds	Allelochemical biocontrol	[65]
Microalgae species	Weeds	Structural diversity of allelochemicals	[75]
*Ulva intestinalis* (formerly *Enteromorpha intestinalis*) (Chlorophyta)	Weeds	Phytotoxic effects	[76]
*Mastocarpus stellatus* and *Porphyra dioica*	white clover (*Trifolium repens* L.) and Italian ryegrass (*Lolium multiflorum*)	Pre- and post-emergence herbicidal effects	[77]
*Plocamium brasiliense* (Rhodophyta)	*Mimosa pudica* and *Senna obtusifolia*	Allelopathic effects halogenated monoterpenes	[79]

Taken together, these results show that seaweeds may have real potential as herbicidal agents. Thanks to the complex chemistry found in these marine plants, especially the way different compounds might interact, they offer a promising and more environmental-friendly alternative to traditional weed killers [73].

## 3. Mechanisms of Action of Algal-Derived Pesticides

Bioactive molecules from marine algae influence a range of biological systems in both pests and pathogens. A notable example is enzyme inhibition. Some species of red algae, for instance *Laurencia obtusa*, *L. nidifica*, *L. johnstonii*, *Palisada perforata* (formerly *L. papillosa*) and *Plocamium cartilagineum* generate halogenated sesquiterpenes and brominated elements that boost reactive oxygen species (ROS) in pest cells [8,23,36,40,44,80,81]. These compounds compete for binding with LuxR-type proteins, ultimately silencing genes involved in virulence and breaking down bacterial coordination [44,82]. This type inhibition of signaling pathways represents a subtle but effective mode of biological control, especially in microorganisms prone to resistance from conventional bactericides. Still, it is hard to achieve consistent results outside the lab since the compounds can break down easily or are not always absorbed well by plants or pests.

Enzyme-targeting compounds are not limited to bacterial systems. Tannins obtained from species of brown algae, including *Sargassum* spp. exhibit strong metal-chelating and protein-binding properties, which inactivate key metalloenzymes involved in respiration and DNA synthesis, thereby disrupting vital processes in more complex eukaryotic pests as well [11,17]. Yet, the effectiveness of these metabolites can fluctuate significantly with changes in harvest timing, algal maturity, and extraction techniques, posing a challenge for standardization in product development [1]. Algae, particularly the green and brown types, are noted for their abundance of fatty acids such as palmitic, oleic, and linoleic [60]. These molecules integrate into pest cell membranes, disturbing their structure. As a result, the membranes become more permeable, leading to fluid imbalance and, in many cases, cell rupture [4,5]. Another important mechanism involves the inhibition of neural enzymes. Phlorotannins isolated from the brown alga *Ecklonia maxima* exhibited significant acetylcholinesterase (AChE) inhibitory activity, providing direct evidence for a neurotoxic mode of action in pest control [83]. Similarly, metabolites from the red alga *Laurencia snackeyi* displayed strong in vitro and in silico cholinesterase inhibition, attributed to halogenated sesquiterpenes [84]. In addition, the cyanobacterial metabolite anatoxin-a(S), produced by *Anabaena flos-aquae*, irreversibly inhibits AChE, leading to overstimulation of synapses and paralysis [85]. Extracts from *Spirulina* sp. and *Nostoc muscorum* reduced the activities of glutathione S-transferase (GST), cytochrome P450, and carboxylesterases in *Spodoptera frugiperda*, thereby impairing the insect’s ability to metabolize and detoxify xenobiotics [86]. Although these agents act swiftly and target a wide range of pests, their persistence in field conditions is limited. Environmental factors—mainly sunlight, temperature, pH, and microbial activity—can accelerate their degradation, shortening their protective effect and making field performance less predictable [24].

Apart from direct toxicity, some marine algal metabolites strengthen plant immunity. Sulfated polysaccharides such as carrageenans [63] and ulvans mimic microbial-associated molecular patterns and stimulate systemic plant defense pathways governed by salicylic acid, jasmonic acid, and ethylene. These processes boost the activity of protective enzymes like chitinase, β-1,3-glucanase, peroxidase, and phenylalanine ammonia-lyase [12,87]. Such treatments offer a useful approach, but their effectiveness often fluctuates as polysaccharide composition changes between algal types and across seasons [42]. Laminarin, extracted from brown algae *Laminaria digitata*, has gained wide recognition in this group for its strong and consistent elicitor effects. It sets off mitogen-activated protein kinase (MAPK) cascades, strengthens ROS signaling, and encourages the production of antimicrobial compounds in plants [47,88].

Nonetheless, the production of highly purified laminarin suitable for agricultural formulations remains both technically complex and cost-intensive. Certain algal metabolites also influence ion transport processes in plant tissues. For instance, some brown algae modulate the activity of H^+^-ATPase pumps in roots and leaves, thereby enhancing nutrient absorption and potentially modifying rhizosphere interactions [65,89]. A review on seaweed fertilizers describes how extracts from brown algae (e.g., *Ascophyllum nodosum*, *Durvillaea antarctica*) and *Kappaphycus alvarezii* (Rhodophyta) improve nutrient uptake (N, P, K, S) in soybean and tomato, enhance plant stress tolerance, and reduce electrolyte leakage—effects often associated with improved ion transport homeostasis [90]. While these adjustments in plant physiology are encouraging for enhancing crop strength and defense, their reliability in the external environment (in actual field settings) still needs to be thoroughly tested [91].

Marine algae’s capability to create hormonal analogs is significant for overseeing plant growth and development. Some of these compounds mimic plant hormones like auxins and cytokinins, influencing root growth, leaf shape, and even the release of volatile organic compounds that can attract helpful insects or ward off pests [43,92,93]. Despite their potential, the intricate interactions and environmental variability keep posing hurdles for dependable application [6]. Algal-origin pesticides stand out because of their remarkable skill in targeting numerous biological mechanisms all at once. They affect different biological pathways—enzyme activity, membrane integrity, and immune responses—making it harder for pests to develop resistance compared to synthetic pesticides that usually target only one site [3,94]. There have been promising outcomes, but regulatory approval is still difficult to obtain. Authorities usually require a well-defined active agent and a clear mode of action, which are hard to determine in naturally diverse algal extracts [14]. Without scalable extraction methods and formulation stabilization, translating this potential into commercially viable products remains a considerable challenge [17,18].

## 4. Methods of Application for Crop Protection

Marine algae can be introduced into agricultural systems through several practical methods, each offering specific advantages for crop protection and productivity. Foliar spraying, in particular, stands out for its efficiency. Applying algae extracts to leaves allows plants to absorb compounds that initiate their own protective responses by activating important signaling pathways [62,95]. After the signals are set off, the plant activates the production of enzymes (polyphenol oxidase and peroxidase) to boost its natural defenses [96].

Empirical evidence suggests that the application of diluted foliar treatments may lower the need for synthetic chemical inputs and improve plant tolerance to stress conditions [51,90]. Recent articles indicated that the foliar application of seaweed formulations significantly aided in the management of early blight in *Solanum lycopersicum* and mitigated the prevalence of peach leaf curl, concurrently contributing to enhanced yield outcomes [97]. The results of these treatments are closely aligned with the extraction method utilized, since methanol or ethanol-based formulations increase the concentration of active phenolics and fatty acids [59,64].

Beyond foliar sprays, soil application of marine algae, whether in composted, powdered, or liquid forms is widely practiced [98]. These applications not only support root system growth and encourage microbial balance but also reduce the occurrence of soil-borne pathogens, including root-knot nematodes and *Fusarium* wilt [48,99]. Red and brown algae, when integrated into soil, allow for slow nutrient release while maintaining pathogen suppression, especially valuable in organic farming systems.

A further successful method is the treatment of seeds, which comprises soaking seeds in diluted algal extracts before planting. This technique is correlated with increased germination rates and seedling vigor, leading to improved resistance to early-stage pathogenic infections [51]. Such seed-focused application prime the seeds by initiating defense-related gene expression and improving internal biochemistry, helping young plants better handle biotic and abiotic challenges. In this regard, the algal species *Kappaphycus alvarezii* (Rhodophyta), *Sargassum polycystum* (formerly *Sargassum myriocystum*) (Phaeophyceae), and *Ulva lactuca*, are particularly effective in supporting early root architecture and vigor [88,100].

Increasingly, marine algal bioproducts are being delivered through integrated methods, including fertigation and root dipping. Multiple delivery methods used in tandem have shown enhanced effectiveness in difficult cultivation conditions. For example, Sahana et al. [101] observed that applying both foliar and soil treatments significantly lowered rice blast severity and increased plant biomass.

Altogether, the flexibility of marine algae in seed priming, foliar spraying, soil conditioning, and integrated treatments positions them as one of the main component of sustainable crop protection strategies. Their compatibility with Integrated Pest Management (IPM) frameworks offers an environmentally sound alternative to synthetic pesticides [90,102]. Future advancements in this domain will rely on refining formulations, establishing consistent dosages, and verifying results under field conditions.

## 5. Advantages of Marine Algae-Based Biopesticides

The basic benefits of marine algae-based biopesticides, such as their safety and pest-selective action, have been covered earlier, so the following section highlights further advantages related to their use and practical challenges that may arise.

From an environmental perspective, marine algae are highly sustainable. They grow quickly, do not need freshwater or farmland, and can be raised in the ocean without causing much harm to the environment. Species from genus *Ulva* (Chlorophyta), *Sargassum* (Phaeophyceae), and *Gracilaria* (Rhodophyta) grow easily in different coastal areas, making them a practical and land-efficient source of biomass [3,103]. Unlike synthetic pesticides, growing algae uses much less energy, produces fewer greenhouse gases, and creates byproducts that support the goals of sustainable and low-carbon agriculture [104,105,106].

Economically, algae-based biopesticides offer advantages that are especially relevant for resource-limited settings. Algal biomass can be locally sourced or cultivated with minimal infrastructure, making it an affordable option for smallholder farmers [107,108]. In many coastal areas, seaweeds commonly collected for food, cosmetics, or fertilizers can also finding new use as raw materials for crop protection products. This not only helps reduce waste but also supports circular economy practices [109]. Their multifunctionality—as pest deterrents, plant growth promoters, and soil conditioners—reduces the need for multiple agricultural inputs, lowering overall production costs [11,110]. Additionally, studies have reported increased yield and quality in treated crops such as tomato, rice, and cucumber, suggesting a strong economic return [48,95]. Algal biopesticides fit naturally into IPM systems and offer a practical solution for sustainable agriculture. Their compatibility with microbial biocontrol agents and minimal impact on non-target organisms further enhances their value in environmentally responsible crop protection [111,112]. Because repeated application of algae-based biopesticides does not promote resistance in pests, they offer a reliable long-term solution that helps to decrease chemical use [113]. Since they are approved for use in organic systems, these products help producers access certified markets and tap into higher-value opportunities [92,102]. Improvement of soil quality is another significant benefit. Applied as compost or soil additives, marine algae supply key nutrients and organic matter that help enhancing soil fertility and structure [114,115]. This creates the right conditions for proliferation of beneficial soil microflora, leading to improved nutrient cycling, stimulate root development, and strengthen plant resilience against abiotic stresses like drought and salinity [99,116]. Such long-term agronomic advantages are generally absent in conventional pest control products, which may instead disrupt microbial balances or degrade soil over time.

Most of the limitations of algae-based biopesticides have already been addressed, including variability in efficacy, challenges with standardization, formulation stability, and supply constraints. Continued research and field testing remain important to overcome these issues.

## 6. Conclusions

The metabolites derived from marine algae are invaluable, possessing various pesticidal capabilities and showcasing a sustainable option relative to traditional agricultural chemicals. The way they merge pest management with encouraging plant growth and supporting soil vitality positions them as essential parts of integrated pest control strategies. Taking lab accomplishments and translating them into credible results on the ground requires refining the stability of formulations, unifying active elements, and crafting tailored application approaches. Collaborative interdisciplinary research and validation in the field will be crucial to fulfilling their potential in robust, low-input agricultural practices.

## 7. Future Directions

Although the study of marine algae has made significant progress, numerous unresolved issues remain. To begin with, cultivating algae specifically for agricultural purposes may appear promising in theory, but it is still uncertain which strains are truly viable for large-scale cultivation. Additional research is necessary to identify the most advantageous species—not only in laboratory settings but also regarding availability and usability for farmers in real-world scenarios.

Furthermore, there is a need to delve deeper into the chemistry of these algal compounds. Some extracts yield positive results, while others do not perform as expected, and the reasons for these discrepancies are not always clear. Investing in advanced tools or interdisciplinary methods that extend beyond conventional testing may be warranted. It is possible that we have been neglecting lesser-known compounds that could play a significant role in efficacy.

Moreover, a broader consideration of integration is essential. Even if marine algae demonstrate effectiveness, how do they integrate with existing agricultural systems? Are they compatible with precision farming technologies, or do they align better with traditional practices? Additionally, how will small-scale farmers react to them in contrast to large-scale producers? These inquiries cannot be resolved in laboratory conditions—they necessitate practical trials and genuine discussions with those involved in food production.

In addition, the impact of climate change introduces an extra layer of urgency. As climatic conditions become more erratic and new pests emerge, algae-based products could offer benefits beyond mere pest management. They may enhance plant resilience to stress or contribute to soil health restoration. We will not uncover these potentials unless we actively investigate them.

All of this underscores one crucial point: future research should not merely reinforce existing knowledge. It must take a step back to pose broader questions—concerning practicality, accessibility, resilience, and the ultimate goals of modern agriculture.

## Figures and Tables

**Table 1 marinedrugs-23-00346-t001:** Insecticidal Activity of Marine Algae.

Algal Species	Target Pest	Product/Effects	Reference
*Sargassum wightii, Turbinaria ornata* (Phaeophyceae)	Cotton leafhopper, *Amrasca biguttula*	Tannins inhibit amylase, protease, invertase	[11]
*Actinotrichia fragilis, Liagora ceranoides* (Rhodophyta), *Sargassum latifolium and Colpomenia sinuosa* (Phaeophyceae)	Stored-grain pests *Oryzaephilus mercator* *Tribolium castaneum*, *Musca domestica*	Chitin synthase inhibition Volatile oils with 39.6% 1-dodecanol insecticidal effects	[25,30]
*Chaetomorpha antennina* (Chlorophyta); *Acanthophora spicifera*, *Gracilaria corticata* and *Jania rubens* (Rhodophyta)	*Spodoptera litura*	Fatty acids induce plant defenses larvicidal and insecticidal effects	[6,31]
*Ulva lactuca* (Chlorophyta)*, Turbinaria ornata* (Phaeophyceae)	*Culex pipiens* larvae	Phenolics, alkaloids cause DNA damage	[17]
*Halymenia dilatata* (Rhodophyta)	*Aedes aegypti*	Midgut damage, enzyme alteration	[32]
*Padina pavonica* (Phaeophyceae)	*Dysdercus cingulatus*	Nymphicidal and ovicidal activity	[26]
*Sargassum tenerrimum* (Phaeophyceae)	*Dysdercus cingulatus*	Reduced protein and DNA content	[33]
Green seaweeds *Caulerpa* and *Ulva* spp. (Chlorophyta)	*Spodoptera litura* *Dysdercus cingulatus* *Drosophila melanogaster*	Larval mortality and growth inhibition, Fatty acid and other metabolite	[29,34,35]
*Laurencia nidifica* (Rhodophyta)	Maize weevil, termites	Halogenated sesquiterpenes inhibit acetylcholinesterase	[36]
*Sargassum wightii*, (Phaeophyceae) *Gracilaria edulis* (Rhodophyta)	Diamondback moth *Plutella xylostella*; rice leaf folder, *Cnaphalocrocis medinalis*	Larvicidal effect contact/feeding toxicity	[27,37]
*Padina tetrastromatica, Sargassum wightii* and *Turbinaria conoides* (Phaeophyceae)	Lepidopteran tobacco cutworm, *Spodoptera litura*	Larvicidal effect contact/feeding toxicity	[38]
*Chaetomorpha antennina* (Chlorophyta)	Lepidopteran tobacco cutworm, *Spodoptera litura*	Larvicidal effect contact/feeding toxicity	[6]
*Fucus spiralis* (Phaeophyceae)	Mediterranean fruit fly, *Ceratitis capitata*	Essential oil contact toxicity on pupae and adults	[22]
*Laurencia johnstonii* (Rhodophyta) and *Sargassum horridum* (Phaeophyceae)	Asian citrus psyllid, *Diaphorina citri*	Insecticidal and repellent effect	[39]
*Palisada perforata* (formerly *Laurencia papillosa*) (Rhodophyta)	Confused flour beetle, *Tribolium confusum*	Active ingredient acetogenin Larvicidal effect	[23]
*Plocamium cartilagineum* (Rhodophyta)	Tomato moth, *Tuta absolute* and a cereal aphid, *Schizaphis graminum*	Halogenated monoterpenes, (mertensene and violacene) larvicidal and insecticidal effects	[40]

**Table 2 marinedrugs-23-00346-t002:** Fungicidal Activity of Marine Algae.

Algal Species	Target Pest	Product/Effects	Reference
*Gracilariopsis persica* (Rhodophyta)	*Botrytis cinerea*, *Aspergillus niger*, *Penicillium expansum*, and *Pyricularia oryzae*	Phenolics, fatty acids inhibit growth	[55]
*Fucus vesiculosus* (Phaeophyceae)	*Fusarium culmorum*	Fatty acids, fucosterol induced defense enzymes in plants	[45]
*Sargassum wightii*	*Sclerotium rolfsii*	Methanol extract retention the antimicrobial activity	[59]
*Halimeda opuntia* *Turbunaria decurrens* (Phaeophyceae)	*A. parasiticus* *A. flavus*	Growth suppression Anti-mycotoxigenic effect	[46]
*Ulva lactuca* (formerly *Ulva fasciata*), *Ulva flexuosa* (formerly *Enteromorpha flexuosa*)	Cucumber fungal pathogens	Growth suppression	[48]
*Osmundea pinnatifida* (Rhodophyta)	*Alternaria infectoria* and *Aspergillus fumigatus*	Fatty acids, *n*-hexane fraction growth suppression and sporulation	[60]
*Sargassum polycystum* (formerly *Sargassum myriocystum*) (Phaeophyceae)	*Macrophomina phaseolina*	Antifungal effects	[61]
*Chaetomorpha antennina* (Chlorophyta)	Fungal phytopathogens	Alginates, laminarins activate plant defenses	[62]
Brown macroalgae	Various fungi	Phlorotannins disrupt fungal membranes	[50]
*Ascophyllum nodosum* (Phaeophyceae)	*Fusarium, Alternaria, Botrytis* spp.	Induced defense enzymes in plants	[12]
*Caulerpa* (Chlorophyta), *Sargassum* (Phaeophyceae), *Gracilaria* (Rhodophyta) spp.	*Macrophomina phaseolina*	Antifungal effect using poison food technique	[61]
*Solieria robusta* (Rhodophyta)	*Fusarium solani*	Suppressive effect	[49]
*Chondracanthus teedei* var. lusitanicus (Rhodophyta)	*Aspergillus fumigatus* and *A. infectoria*	Carrageenans growth suppression	[63]
*Colpomenia sinuosa*, *Padina pavonia*, *Gongolaria barbata* (formerly *Cystoseira barbata*) and *Sargassum vulgare* (Phaeophyceae)	*Aspergillus niger*, *A. flavus*, *Penicillium parasiticus* and *F. solani*	Polar and nonpolar extracts antifungal effects	[64]

**Table 3 marinedrugs-23-00346-t003:** Nematicidal Activity of Marine Algae.

Algal Species	Target Pest	Product/Effects	Reference
Multiple Baja California seaweeds	*Meloidogyne incognita*	Phenolics with nematicidal activity	[44]
*Phacelocarpus tristichus, Turbinaria ornata*	*Meloidogyne incognita*	Suppressed nematodes, promoted plant growth	[4]
*Ulva lactuca* (*formerly Ulva fasciata*), *Corallina* spp. (Rhodophyta)*, Limnospira platensis* (formerly *Spirulina platensis*) (Cyanobacteria)	*Meloidogyne incognita*	Reduced egg hatching, larval survival	[71]
Chlorophyta (green microalgae)	Model organisms	Fatty acids, alkaloids show nematicidal effects	[5]
32 seaweed species	*Meloidogyne javanica*	Inhibited egg hatching, larval mortality	[72]
Algae and cyanobacteria	Plant-parasitic nematodes	Suppression through bioactive compound release	[66]
*Ascophyllum nodosum* and *Ecklonia maxima* (Phaeophyceae)	*Meloidogyne chitwoodi* and *Meloidogyne hapla*	Suppressed nematodes, promoted plant growth	[70]
*Solieria robusta* (Rhodophyta)	*Meloidogyne javanica*	Reduced egg hatching and nematode viability	[49]
*Colpomenia sinuosa* (Phaeophyceae) *Corallina mediterranea* (Rhodophyta, Corallinaceae)	*Meloidogyne incognita*	Nanoparticle formulations nematicidal activity	[67]

## Data Availability

Not applicable.

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
