# Peer review of "Harnessing Marine Algae for Sustainable Agriculture: Natural Bioactive Compounds as Eco-Friendly Pesticidal Agents"

_marinedrugs, 2025, doi:10.3390/md23090346_

Round 1

Reviewer 1 Report

Comments and Suggestions for Authors This paper provides a relatively detailed review of research on marine algae as eco-friendly pesticides. Here are some suggestions:

  1. In the "Pesticidal Properties of Marine Algae" section, the authors list the activities of components such as fatty acids and tannins, or methanol or ethanol extracts . Please present studies on the activities of specific compounds in the literature, it is recommended to list the names and structural types of the active compounds.
  2. In the "Mechanisms of Action of Algal-Derived Pesticides" section, the authors mention mechanisms such as disrupting cell structure, inhibiting enzyme activity, interfering with metabolic processes, and enhancing plant immunity. It is suggested that this section be further explored in depth. For example, are there reports on the inhibition of metabolic detoxifying enzymes? Do the Alpal-derived pesticides have an impact on acetylcholinesterase, glutathione S-transferase, etc.?
  3. Regarding the application of marine algae, the authors mainly discuss their protective effects during the plant growth stage. Can they be used for the control of stored-grain pests?

Author Response

Response to Reviewer Comments

We sincerely thank the reviewer for the constructive suggestions. Our detailed responses are provided below with all revisions highlighted in red in the revised version.

Comment 1: In the "Pesticidal Properties of Marine Algae" section, the authors list the activities of components such as fatty acids and tannins, or methanol or ethanol extracts. Please present studies on the activities of specific compounds in the literature, it is recommended to list the names and structural types of the active compounds.

Response 1: Thank you for raising this important point. In the original version, much of our discussion emphasized crude extracts, and we agree it is more informative to highlight specific compounds and their structural types. We therefore expanded this section with examples where the active metabolites have been identified. For instance, Sargassum wightii was reported to contain ethyl salicylate, methyl salicylate, ethyl palmitate, palmitic acid, oleic acid, phytolin, and diethyl phthalate, which contribute to its larvicidal activity (Lines 83–85). Similarly, Chaetomorpha antennina produces fatty acids such as hexadecanoic, octadecanoic, and linoleic acids that trigger defense responses in tomato plants against Spodoptera litura (Lines 88–90). Volatile oils from Actinotrichia fragilis, Liagora ceranoides, and Colpomenia sinuosa caused substantial mortality in Oryzaephilus mercator and Tribolium castaneum, with the oil of A. fragilis, rich in 1-dodecanol, achieving the highest effectiveness (Lines 91–97). Extracts of Sargassum latifolium were also reported to damage insect exoskeletons and likely act as chitin synthase inhibitors, as supported by molecular docking studies (Lines 97–103). Additional examples were incorporated for antifungal activity, including norcycloartane-type triterpenes, phlorotannins, rosmarinic acid, quercetin, and sterols (Lines 138–141; 143-150), as well as nematicidal compounds identified in Colpomenia sinuosa (Lines 176–184). These additions ensure the section emphasizes compound-level insights rather than relying solely on general extract activity.

Comment 2: In the "Mechanisms of Action of Algal-Derived Pesticides" section, the authors mention mechanisms such as disrupting cell structure, inhibiting enzyme activity, interfering with metabolic processes, and enhancing plant immunity. It is suggested that this section be further explored in depth. For example, are there reports on the inhibition of metabolic detoxifying enzymes? Do the Algal-derived pesticides have an impact on acetylcholinesterase, glutathione S-transferase, etc.?

Response 2: We appreciate this suggestion and have broadened the discussion on specific enzyme targets. New evidence has been added to describe the inhibition of neural enzymes, including phlorotannins from Ecklonia maxima, which significantly inhibited acetylcholinesterase (AChE) (Lines 271–273), and halogenated sesquiterpenes from Laurencia snackeyi, which also showed strong cholinesterase inhibition both in vitro and in silico (Lines 274–276). The cyanobacterial metabolite anatoxin-a(S), produced by Anabaena flos-aquae, was also included as it irreversibly inhibits AChE, resulting in paralysis (Lines 277–278). Alongside their effects on neural pathways, we also emphasized how algal metabolites interfere with key detoxification enzymes. Extracts from Spirulina sp. and Nostoc muscorum were shown to reduce the activity of glutathione S-transferase (GST), cytochrome P450, and carboxylesterases in Spodoptera frugiperda, thereby weakening the insect’s ability to metabolize and detoxify xenobiotics (Lines 278–281). Together, these examples provide a strong mechanistic framework by connecting algal metabolites to both neurotoxic and detoxification-inhibiting actions.

Comment 3: Regarding the application of marine algae, the authors mainly discuss their protective effects during the plant growth stage. Can they be used for the control of stored-grain pests?

Response 3: This is an excellent point. To address it, we expanded the section to highlight evidence that marine algae can also be used against stored-product pests. Volatile oils from Actinotrichia fragilis, Liagora ceranoides, and Colpomenia sinuosa caused substantial mortality in Oryzaephilus mercator and Tribolium castaneum, with the oil of A. fragilis, rich in 1-dodecanol, achieving the highest effectiveness (Lines 91–97). Extracts of Sargassum latifolium were also reported to damage insect exoskeletons and likely act as chitin synthase inhibitors, as supported by molecular docking studies (Lines 97–103). In addition to insect pests, we highlighted content already present in the manuscript showing that Auxenochlorella protothecoides suppressed postharvest fungal pathogens (Lines 153–157). These findings together demonstrate that marine algal metabolites have potential not only for crop protection during growth but also as eco-friendly alternatives in stored-grain management.

Reviewer 2 Report

Comments and Suggestions for Authors

The manuscript titled "Harnessing Marine Algae for Sustainable Agriculture: Natural Bioactive Compounds as Eco-Friendly Pesticidal Agents" addresses a relevant topic and is appropriate for this journal.

Overall, the manuscript is well-structured and well-researched.

However, the manuscript requires a thorough review of the taxonomy/nomenclature of the algae cited, as indicated below.

Corrections needed:

line 74 - develop properly—or survive at all [25]. The species Padina pavonica (Phaeophyceae) also shows good in-

Table 1 - Chaetomorpha antennina (Chlorophyta); Acanthophora spicifera, Gracilaria corticata and Jania rubens(Rhodophyta)

Ulva lactuca (Chlorophyta), Turbinaria ornata (Phaeophyceae)

Halymenia dilatata (Rhodophyta)

Padina pavonica (Phaeophyceae)

Sargassum tenerrimum (Phaeophyceae)

Caulerpa and Ulva spp. (Chlorophyta)  (Note: "spp." is not in italics)

Laurencia nidifica (Rhodophyta)

Sargassum wightii (Phaeophyceae), Gracilaria edulis (Rhodophyta)

Padina tetrastromatica, Sargassum wightii and Turbinaria conoides (Phaeophyceae)

Chaetomorpha antennina (Chlorophyta)

Fucus spiralis (Phaeophyceae)

Laurencia johnstonii (Rhodophyta) and Sargassum horridum (Phaeophyceae)

Palisada perforata (formerly Laurencia papillosa) (Rhodophyta)

Plocamium cartilagineum (Rhodophyta)

line 104 - Ulva lactuca (formerly Ulva fasciata) and Ulva flexuosa (formerly Enteromorpha flexuosa) (Chlorophyta), two green algae, reduced the growth of soil

line 106 - of Ulva lactuca (formerly U. fasciata) cut fungal proliferation by nearly 69% [37]. The red seaweed Solieria robusta,

line 108 - suppressed nematodes [38]. Extracts from Laurencia johnstonii (Rhodophyta) and Dictyota dichotoma (Phaeophyceae)), rep- 

line 110 - gens [33, 39]. Other brown species like Cystoseira humilis var. myriophylloides (formerly Cystoseira myriophylloides) and Laminaria digitata (Phaeophyceae) lim-

line 116 - from Ascophyllum nodosum (Phaeophyceae) enhanced chitinase and β-1,3-glucanase in cucumber, leading 

lines 1207/121 - mented effects against Botrytis and Curvularia spp.) [39]. Meanwhile, Auxenochlorella protothecoides (formerly Chlorella protothecoides), a well-known freshwater green microalga (some strains can tolerate brackish conditions)

Table 2 - Gracilariopsis persica (Rhodophyta)

Fucus vesiculosus (Phaeophyceae)

Sargassum wightii (Phaeophyceae)

Halimeda opuntia (Phaeophyceae) 

Turbunaria decurrens (Phaeophyceae)

Ulva lactuca (formerly Ulva fasciata), Ulva flexuosa (formerly Enteromorpha flexuosa)

Osmundea pinnatifida (Rhodophyta)

Sargassum polycystum (formerly Sargassum myriocystum) (Phaeophyceae)

Chaetomorpha antennina (Chlorophyta)

Ascophyllum nodosum (Phaeophyceae)

Caulerpa (Chlorophyta), Sargassum (Phaeophyceae), Gracilaria (Rhodophyta) spp. (Note: "spp" is not in italics)

Solieria robusta (Rhodophyta)

Chondracanthus teedei var. lusitanicus (Rhodophyta)

Colpomenia sinuosa, Padina pavonica,  Gongolaria barbata (formerly Cystoseira barbata) and Sargassum vulgare (Phaeophyceae)

tions in nematode populations and enhanced plant biomass. The seaweed included green

line 145 - dered seaweed to nematode-infested soils or tomato plant roots led to significant reduc-

line 155 - brown algae species Phacelocarpus tristichus, Turbinaria ornata, Ascophyllum nodosum and

line 158 - 56]. Ethanolic extracts of Padina concrescens (Phaeophyceae) and Laurencia johnstonii (Rhodophyta) also stopped a large

line 160 - while, soil treatments using Spatoglossum schroederi (Phaeophyceae) cut down root galling and improved

Table 3 - Ulva lactuca (formerly Ulva fasciata) (Chlorophyta), Corallina spp. (Rhodophyta), Limnospira platensis (formerly Spirulina platensis) (Cyanobacteria)

Algae and Cyanobacteria 

Ascophyllum nodosum and Ecklonia maxima (Phaeophyceae)

Solieria robusta (Rhodophyta)

line 186 - lettuce seed growth. Seven of them had a noticeable impact at 1,000 ppm. Notably, Laurencia heteroclada (Rhodophyta), Caulerpa racemosa, and Caulerpa sertularioides (Chlorophyta) were the most effective in 

Table 4 - Ulva intestinalis (formerly Enteromorpha intestinalis) (Chlorophyta)

Mastocarpus stellatus and Porphyra dioica (Rhodophyta)

Plocamium brasiliense (Rhodophyta)

line 206 - for instance Laurencia obtusa, L. nidifica, L. johnstonii, Palisada perforata (formerly L. papillosa) and Plocamium cartilagi-

line 216 - from species of brown algae, including Sargassum spp. exhibit strong metal-chelating and (Note: "spp." is not in italics)

line 236 - inarin, extracted from brown alga Laminaria digitata, has gained wide recognition in this 

line 245 - seaweed fertilizers describes how extracts from brown and red algae (e.g. Ascophyllum nodosum, 

line 294 - handle biotic and abiotic challenges. In this regard, the algal species Kappaphycus alvarezii (Rhodophyta), 

line 295 - Sargassum polycystum (formerly Sargassum myriocystum) (Phaeophyceae), and Ulva lactuca (Chlorophyta), are particularly effective in supporting early

line 314 - ing much harm to the environment. Species from genus Ulva (Chlorophyta), Sargassum (Phaeophyceae), and Gracilaria (Rhodophyta)

Author Response

Response to Reviewer Comments

We thank the reviewer for their valuable suggestions regarding taxonomy and nomenclature. We have carefully reviewed the entire manuscript, ensuring that all algal species names are correctly italicized and updated to reflect current taxonomy including the new information added. The corresponding higher taxonomic group (Chlorophyta, Rhodophyta, or Phaeophyceae) is specified after the scientific name at its first appearance only. When a sentence already refers to “green algae,” “red algae,” or “brown algae,” we have intentionally omitted the Latin group designation to avoid tautology and unnecessary repetition.

All corrections have been implemented in the revised manuscript. Below, we provide point-by-point responses with all revisions highlighted in red in the revised version.

  1. Former line 74 (now lines 74):
  • Corrected as: “Padina pavonica (Phaeophyceae) also shows good insecticidal activity…”
  1. Table 1 (Insecticidal Activity of Marine Algae):
  • All names corrected and italicized. “spp.” consistently uprighted.
  • First appearance includes group name; subsequent entries omit group name when redundant.
  • Chaetomorpha antennina (Chlorophyta); Acanthophora spicifera, Gracilaria corticata, Jania rubens (Rhodophyta); Ulva lactuca (Chlorophyta); Padina pavonica (Phaeophyceae); Palisada perforata (formerly Laurencia papillosa) (Rhodophyta).
  1. Former lines 104–116 (now 125–141):
  • Updated nomenclature with group designation at first mention only:
    • Ulva lactuca (formerly Ulva fasciata) (Chlorophyta)
    • Ulva flexuosa (formerly Enteromorpha flexuosa) (Chlorophyta)
    • Laurencia johnstonii (Rhodophyta)
    • Dictyota dichotoma (Phaeophyceae)
    • Cystoseira humilis var. myriophylloides (formerly Cystoseira myriophylloides) (Phaeophyceae)
    • Laminaria digitata (Phaeophyceae)
    • Ascophyllum nodosum (Phaeophyceae)
  1. Former lines 1207/121 (now 153-154):
  • Corrected: Auxenochlorella protothecoides (formerly Chlorella protothecoides) (Chlorophyta).
  1. Table 2 (Fungicidal Activity of Marine Algae):
  • All names updated with same formatting principle.
    • Gracilariopsis persica (Rhodophyta)
    • Fucus vesiculosus (Phaeophyceae)
    • Ulva lactuca (formerly U. fasciata) (Chlorophyta)
    • Ulva flexuosa (formerly Enteromorpha flexuosa) (Chlorophyta)
    • Sargassum polycystum (formerly Sargassum myriocystum) (Phaeophyceae)
    • Chondracanthus teedei var. lusitanicus (Rhodophyta)
    • Gongolaria barbata (formerly Cystoseira barbata) (Phaeophyceae)
  1. Former lines 145–160 (now 189–203):
  • Names corrected and groups added at first mention: Halimeda tuna (Chlorophyta), Spatoglossum asperum, Sargassum swartzii, Spatoglossum variabile (Phaeophyceae), Melanothamnus afaqhusainii (Rhodophyta), and Spatoglossum schroederi (Phaeophyceae).
  1. Table 3 (Nematicidal Activity of Marine Algae):
  • Names corrected and italicized; group indicated only at first mention.
    • Ulva lactuca (formerly U. fasciata) (Chlorophyta)
    • Corallina spp. (Rhodophyta)
    • Limnospira platensis (formerly Spirulina platensis) (Cyanobacteria)
    • Ascophyllum nodosum (Phaeophyceae)
    • Ecklonia maxima (Phaeophyceae)
    • Solieria robusta (Rhodophyta)
    • Colpomenia sinuosa (Phaeophyceae)
  1. Former lines 181–187 (now 225–232):
  • Corrected species names: Mastocarpus stellatus and Porphyra dioica (Rhodophyta), Laurencia heteroclada (Rhodophyta), Caulerpa racemosa (Chlorophyta), Caulerpa sertularioides (Chlorophyta),

Table 4. Herbicidal Activity of Marine Algae

  • Names corrected

Ulva intestinalis (formerly Enteromorpha intestinalis) (Chlorophyta), Plocamium brasiliense (Rhodophyta).

  1. Former lines 206 (now 252):
  • Names corrected

Palisada perforata (formerly L. papillosa)

10. Former lines 246 (now 302)

  • Names corrected

Kappaphycus alvarezii (Rhodophyta)

11. Former lines 294-295 (now 350-351)

  • Names corrected

Kappaphycus alvarezii (Rhodophyta), Sargassum polycystum (formerly Sargassum myriocystum) (Phaeophyceae)

Round 2

Reviewer 1 Report

Comments and Suggestions for Authors

The paper has made the necessary revisions to the issues raised previously.